# Koopman Kernel Regression

**Petar Bevanda**
TU Munich
petar.bevanda@tum.de

**Max Beier**
TU Munich
max.beier@tum.de

**Armin Lederer**
TU Munich
armin.lederer@tum.de

**Stefan Sosnowski**
TU Munich
sosnowski@tum.de

**Eyke Hüllermeier**
LMU Munich
eyke@ifi.lmu.de

**Sandra Hirche**
TU Munich
hirche@tum.de

## Abstract

Many machine learning approaches for decision making, such as reinforcement learning, rely on simulators or predictive models to forecast the time-evolution of quantities of interest, e.g., the state of an agent or the reward of a policy. Forecasts of such complex phenomena are commonly described by highly nonlinear dynamical systems, making their use in optimization-based decision-making challenging. Koopman operator theory offers a beneficial paradigm for addressing this problem by characterizing forecasts via linear time-invariant (LTI) ODEs, turning multi-step forecasts into sparse matrix multiplication. Though there exists a variety of learning approaches, they usually lack crucial learning-theoretic guarantees, making the behavior of the obtained models with increasing data and dimensionality unclear. We address the aforementioned by deriving a universal Koopman-invariant reproducing kernel Hilbert space (RKHS) that solely spans transformations into LTI dynamical systems. The resulting *Koopman Kernel Regression (KKR)* framework enables the use of statistical learning tools from function approximation for novel convergence results and generalization error bounds under weaker assumptions than existing work. Our experiments demonstrate superior forecasting performance compared to Koopman operator and sequential data predictors in RKHS.

## 1 Introduction

Dynamical systems theory is a fundamental paradigm for understanding and modeling the time evolution of a phenomenon governed by certain underlying laws. Such a perspective has been successful in describing countless real-world phenomena, ranging from engineering mechanics [1] and human movement modeling [2] to molecular and quantum systems [3, 4]. However, as the laws governing dynamical systems are often unknown, modeling and understanding the underlying phenomena may have to rely on data rather than first principles. In this regard, machine learning methods, which have shown immense potential in tackling complex tasks in domains such as language models [5] and computer vision [6], are coming to the fore. Though powerful, state-of-the-art neural vector fields [7] or flows [8] commonly compose highly nonlinear maps for forecast, i.e. computing

$$x(t) = x(0) + \int_0^t f(x(t)) \mathrm{d}t \tag{1}$$

for, e.g. a scalar ODE $\dot{x} = f(x)$. Hence, it is often challenging to use such models in optimization-based decision making that relies on simulators or predictive models, e.g., reinforcement learning [9–11]. A particularly beneficial perspective for dealing with the aforementioned comes from Koopman operator theory [12–15]. Through a point-spectral decomposition of Koopman operators, forecasts become superpositions of solution curves of a set of linear ODEs $\{\dot{z}_j = \lambda_j z_j\}_{j=1}^D$

$$x(t) = \sum_{k=1}^D \mathrm{e}^{\lambda_j t} z_j(0), \qquad \{x \overset{\text{lift } g_j}{\longmapsto} z_j\}_{j=1}^D \tag{2}$$

37th Conference on Neural Information Processing Systems (NeurIPS 2023).

where a vector valued function $\text{span}(\{g_j\}_{j=1}^D)$ "lifts" $x$ onto a manifold $\mathbb{Z} := \text{span}\left(\{z_j\}_{j=1}^D\right)$. Throughout, we refer to these models as *linear time-invariant (LTI) predictors*. The learning objective of such representations is twofold: spanning system trajectories by the learned manifold $\mathbb{Z}$ and constraining the LTI dynamics to it. The latter is a long-standing challenge of Koopmanism [16–20], as manifold dynamics of existing approaches "leak-out" [21] and limit predictive performance.

To tackle the aforesaid, we connect the representation theories of reproducing kernel Hilbert spaces (RKHS) and Koopman operators. As a first in the literature, we derive a universal kernel whose RKHS exclusively spans manifolds invariant under the dynamics, as depicted in Figure 1. A key corollary of

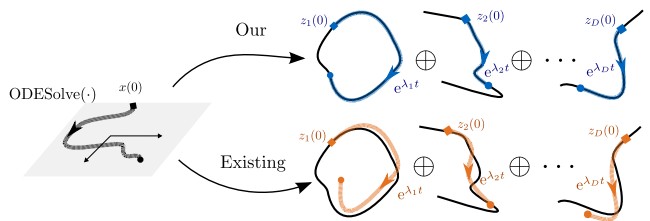

Figure 1: Illustration of on-manifold dynamics of LTI predictors.

unconstrained manifold dynamics is the lack of essential learning-theoretic guarantees, making the behavior of existing learned models unclear for increasing data and dimensionality. To address this, we utilize equivalences to function regression in RKHS to formalize a statistical learning framework for learning LTI predictors from sample trajectories of a dynamical system. This, in turn, enables the use of statistical learning tools from function approximation for novel convergence results and generalization error bounds under weaker assumptions than before [22–24]. Thus, we believe that our Koopman Kernel Regression (KKR) framework takes the best of both RKHS and Koopmanism by leveraging modular kernel learning tools to build provably effective LTI predictors.

The remainder of this paper is structured as follows: We briefly introduce LTI predictors and discuss related work in Section 2. The derivation of the KKR framework, including the novel Koopman RKHS, is presented in Section 3. In Section 4, we show the novel learning guarantees in terms of convergence and generalization error bounds. They are validated in comparison to the state-of-the-art through numerical experiments in Section 5.

**Notation** Lower/upper case bold symbols $x/X$ denote spatial vector/matrix-valued quantities. A *trajectory* defines a curve $x_T \subset \mathbb{X}$ traced out by the flow over time $\mathbb{T} = [0, T]$ from any $(\tau, x) \in \mathbb{T} \times \mathbb{X}$. In discretizing $\mathbb{T}$, collection of points $x_H \subset \mathbb{X}$ from discrete time steps $\mathbb{H} = \{t_0 \cdots t_H\}$ is considered. The state/output trajectory spaces are denoted as $\mathbb{X}_T \subseteq L^2(\mathbb{T}, \mathbb{X})$ / $\mathbb{Y}_T \subseteq L^2(\mathbb{T}, \mathbb{Y})$, with discrete-time analogues $\mathbb{X}_H \subseteq \ell^2(\mathbb{H}, \mathbb{X})$ / $\mathbb{Y}_H \subseteq \ell^2(\mathbb{H}, \mathbb{Y})$ with domain and co-domain separated by ",". The vector space of continuous functions on $\mathbb{X}_T$ endowed with the topology of uniform convergence on compact domain subsets is denoted $C(\mathbb{X}_T)$. The collection of bounded linear operators from $\mathbb{Y}_T$ to $\mathbb{Y}_T$ is denoted as $\mathcal{B}(\mathbb{Y}_T)$. The adjoint of $\mathcal{A} \in \mathcal{B}(\cdot)$ is $\mathcal{A}^*$. Discrete-time eigenvalues read $\mu := e^{\lambda \Delta t}, \lambda \in \mathbb{C}$. A random variable $X$ defined on a probability space $(\Omega, \mathcal{A}, \rho)$ has expectation $\mathbb{E}[X] = \int_\Omega X(\omega)\rho(\omega)$.

## 2 Problem Statement and Related Work

To begin, we formalize our problem statement and put our work into into context with existing work.

### 2.1 Problem Statement

Consider a forward-complete system[1] comprising a nonlinear state-space model

$$\dot{x} = f(x), \ x_0 = x(0), \tag{3a}$$

$$y = q(x), \tag{3b}$$

on a compact domain $\mathbb{X} \subset \mathbb{R}^d$ with a quantity of interest $q \in C(\mathbb{X})$. The above system class includes all systems with Lipschitz flow $F^t(x_0) := \int_0^t f(x(\tau))d\tau$, e.g., mechanical systems [27].

Inspired by the spectral decomposition of Koopman operators, we look to replace the nonlinear state-space model (3) by an *LTI predictor*

$$\dot{z} = Az, \ z_0 = g(x_0), \tag{4a}$$

$$y = c^\top z, \tag{4b}$$

---

[1]Although we outline the scalar output case for ease of exposition, expanding to a vector-valued case is possible w.l.o.g. If required, forward completeness can be relaxed to unboundedness observability [25, 26].

with $\bar{D} \in \mathbb{N}$ and $\boldsymbol{g}$ a $\bar{D}$-dimensional function approximator dense in $C(\mathbb{X})$. Then, from initial conditions $\mathbb{X}_0 \subseteq \mathbb{X}$ that form a *non-recurrent* domain $\mathbb{X}_T$, (4) admits a universal approximation of the flow of (3) such that $\forall \varepsilon > 0$, $\exists \bar{D}$ so that $\sup_{\boldsymbol{x} \in \mathbb{X}_0} |y_T(t) - \boldsymbol{c}^\top \mathrm{e}^{\boldsymbol{A}t} \boldsymbol{g}(\boldsymbol{x})| < \varepsilon$, $\forall t \in [0, T]$ [28][2]. In this work, we aim to find a solution to the following constrained, functional optimization problem

**(OR)** *Output reconstruction:*

$$\min_{\boldsymbol{c}, \boldsymbol{g}, \boldsymbol{A}} \|y_T - \boldsymbol{c}^\top \boldsymbol{g}_T\|_{\mathbb{Y}_T}, \tag{5a}$$

**(KI)** *Koopman-invariance:*

$$\text{such that} \quad \boldsymbol{g}(\boldsymbol{x}(t)) = \mathrm{e}^{\boldsymbol{A}t} \boldsymbol{g}(\boldsymbol{x}(0)), \qquad \forall t \in [0, T]. \tag{5b}$$

Although the sought-out model (4) is simple, the above problem is non-trivial and much of the existing body of work utilizes different simplifications that often lead to undesirable properties. In the following, we elaborate on these properties and motivate our novel sample-based solution to (5), which remains relatively simple but nonetheless ensures a well-defined solution with strong learning guarantees.

## 2.2 Related Work

**Koopman operator regression in RKHS** Equipped with a rich set of estimators, operator regression in RKHS seeks a sampled-data solution to

$$\min_{\boldsymbol{A}} \|\boldsymbol{g}(\boldsymbol{x}(t)) - \boldsymbol{A}^*\boldsymbol{g}(\boldsymbol{x}(0))\|_{L^2}, \tag{6}$$

with $\boldsymbol{A}$ a Hilbert-Schmid operator [29] — commonly known as KRR, and EDMD (PCR) or RRR when under different fixed-rank constraints [23, 24]. The choice of RKHS $\boldsymbol{g}$ is commonly one that is dense in a suitable $L^2$ space, e.g. that of the RBF kernel. By an additional projection, a quantity of interest can be predicted via a mode decomposition of the estimated operator, leading to a model akin to (4). In the light of (5), the feature map $\boldsymbol{g}$ is predetermined while violating **(KI)** is merely minimized for a single time-instant $t$. As a consequence, such approaches are oblivious to the time-series structure — offering limited predictive power over the time interval $[0, T]$ of a trajectory as displayed in Figure 1. The extent to which **(KI)** is violated due to spectral properties [30] or estimator bias [24] is known as spectral pollution [31]. The strong implications of this phenomena, motivate regularization [32] and spectral bias measures [24] to reduce its effects. Due to the above challenges, guarantees for Koopman operator regression (KOR) have only recently gained increased attention. Often, however, existing theoretical results [29, 33] are generally not applicable to nonlinear dynamics [34] due to the commonly unavoidable misspecification [35] of the problem (6) incurred by neglecting **(KI)**. The first more general statistical learning results [23, 24] are derived in a stochastic setting under the assumption that the underlying operator is compact and self-adjoint. In stark contrast, the same set of assumptions is restrictive for the deterministic setting [35]: compactness only holds for affine deterministic dynamics [36, 37] while self-adjointness is known to generally not hold for Koopman operators [13, 38, 39]. Regardless of the setting, however, the state-of-the-art exhibits alarming properties: forecasting error not necessarily vanishing with LTI predictor (4) rank [23, Theorem 1] and risk based on a single time-instant.

**Learning via Koopman eigenspaces** Geared towards LTI predictors and closer to our own problem setting (2), another distinct family of approaches aims to directly learn the operator's invariant subspaces [28, 40–42]. The goal is to fit $\boldsymbol{g}(\cdot)$ based on approximate Koopman operator eigenfunctions that still fit the output of interest **(OR)**. However, existing data-driven approaches in this line of work rely on ad-hoc choices and lack essential learning-theoretic properties such as feasibility and uniqueness of solutions — prohibiting provably accurate and automated LTI predictor learning.

**Kernels for sequential data** Motivated by the lack of priors that naturally incorporate streaming and sequential data, there is an increasing interest in *signatures* [43]. They draw from the rich theory of controlled differential equations (CDEs) [44, 45] and build models that depend on a time-varying observation history. An RKHS suitable for sequence modeling is induced by a signature transformation of a base/static RKHS. Generally, if the latter is universal, so are the signature kernels [46]. While arguably more general and well-versed for discriminative and generative tasks [47], forecasting using signature kernels [48] comes at a price, as their nonlinear dependence on observation streams leads to a significant complexity increase compared to LTI predictors.

---

[2]Background on prerequisite Koopman operator theory can be found in the supplementary material.

Motivated by the restrictions of existing Koopman-based predictors, we propose a *function approximation* approach that exploits exploits time-series data and Koopman operator theory to provably learn LTI predictors. Through a novel *invariance transform* we can satisfy **(KI)** by construction and directly minimize the forecasting risk over an entire time-interval **(OR)**. In simple terms: Koopman operator regression fixes $g(\cdot)$ and regresses $\boldsymbol{A}$ and $\boldsymbol{c}$ in (5), whereas our KKR approach selects $\boldsymbol{A}$ to jointly regress $\boldsymbol{c}$ and $g(\cdot)$. Similar in spirit to generalized Laplace analysis [21, 49], our approach allows the construction of eigenmodes from data without inferring the operator itself. Crucially, we demonstrate that selecting $\boldsymbol{A}$ requires no prior knowledge as confirmed by our theoretical results and experiments. To facilitate learning LTI predictors, we derive *universal* RKHSs that are *guaranteed* to satisfy **(KI)** over trajectories — a first in the literature. The resulting equivalences to function regression in RKHS allow for more general and complete learning guarantees in terms of consistency and risk bounds that are free of restrictive operator-theoretic assumptions.

## 3 Koopman Kernel Regression

With the optimization (5) being prohibitively hard due to nonlinear and possibly high dimensional constraints, we eliminate the constraints (5b) by enforcing the feature map $g(\cdot)$ to have the dynamics of intrinsic LTI coordinates associated with Koopman operators, i.e., their (open) eigenfunctions [21].

**Definition 1.** *A Koopman eigenfunction $\phi_{\lambda \in \mathbb{C}} \in C(\mathbb{X})$ satisfies $\phi_\lambda(\boldsymbol{x}) = \mathrm{e}^{-\lambda t} \phi_\lambda(\boldsymbol{F}^t(\boldsymbol{x})), \forall t \in [0, T]$.*

It is proven that Koopman eigenfunctions from Definition 1 are universal approximators of continuous functions [28] — making them a viable replacement for the feature map $g(\cdot)$ in (4). However, following their definition, it is evident that Koopman eigenfunctions are by no means arbitrary due to their inherent dependence on the dynamics' flow. Using the well-established fact that Koopman operators compose a function with the flow, i.e., $\mathcal{K}^t g(\cdot) = g(\boldsymbol{F}^t(\cdot))$, it becomes evident the eigenfunctions from Definition 1 are (semi)group invariants, as they remain unchanged after applying $\{\mathrm{e}^{-\lambda t} \mathcal{K}^t\}_{t=0}^T$. Thus, inspired by the seminal work of Hurwitz on constructing invariants [50], we can equivalently reformulate (5) as an unconstrained problem and jointly optimize over eigenfunctions[3].

**Lemma 1** (Invariance transform). *Consider a function $g \in C(\mathbb{X}_0)$ over a set of initial conditions $\mathbb{X}_0 \subseteq \mathbb{X}$ that form a non-recurrent domain $\mathbb{X}_T$. The invariance transform $\mathcal{I}_\lambda^T$ transforms $g$ into an Koopman eigenfunction $\phi_\lambda \in C(\mathbb{X}_T)$ for (3a) with LTI dynamics described by $\lambda \in \mathbb{C}$*

$$\phi_\lambda(\boldsymbol{x}_T) = \mathcal{I}_\lambda^T g(\boldsymbol{x}_0) := \int_{\tau=0}^T \mathrm{e}^{-\lambda(\tau-t)} g(\boldsymbol{F}^\tau(\boldsymbol{x}_0)) d\tau. \tag{7}$$

The above Lemma 1 is a key stepping stone towards deriving a representer theorem for LTI predictors. However, it is also interesting in its own right as it provides an explicit expression for the flow of an eigenfunction from any point in the state space. Thus, it provides a recipe to obtain a function space that fulfills **(KI)** by construction. As we show in the following, a sufficiently rich set of eigenvalues [28] and Lemma 1 will allow for a reformulation of (5) into an unconstrained problem

$$\min_M \|y_T - M(\boldsymbol{x}_T)\|_{\mathbb{Y}_T}. \tag{8}$$

where the operator $M(\cdot) := \boldsymbol{1}^\top [\phi_{\lambda_1}(\cdot) \cdots \phi_{\lambda_{\bar{D}}}(\cdot)]^\top$ is universal and consisting of Koopman-invariant functions.

### 3.1 Functional Regression Problem

Notice that the problem reformulation (8) is still intractable, as a closed-form expression for the flow map is generally unavailable even for known ODEs. This requires integration schemes that can introduce inaccuracies over a time interval $[0, T]$. Thus, to make the above optimization problem tractable, data samples are used — ubiquitous in learning dynamical systems.

**Assumption 1.** *A collection of $N$ pairs of trajectories $\mathbb{D}_N = \{\boldsymbol{x}_T^{(i)}, y_T^{(i)}\}_{i=1}^N \in (\mathbb{X}_T \times \mathbb{Y}_T)^N$ is available.*

By aggregating different invariance transformations (7) into the *mode decomposition operator*

$$M(\cdot) \equiv \sum_{j=1}^{\bar{D}} \phi_{\lambda_j}(\cdot) : \ \mathbb{X}_T \mapsto \mathbb{Y}_T, \tag{9}$$

---

[3]Proofs for all theoretical results can be found in the supplementary material.

we can formulate a supervised learning approach in the following.

**Learning Problem** With Assumption 1 and Lemma 1, the sample-based approximation of problem (8) reduces to solving

$$\min_M \sum_{i=1}^{N} \|y_T^{(i)} - M(\boldsymbol{x}_T^{(i)})\|_{\mathbb{Y}_T}. \tag{10}$$

while preserving the mode decomposition structure (9). To realize the above learning problem, we resort to the theory of reproducing kernels [51, 52] and look for an operator $\hat{M} \in \mathcal{H}$, where $\mathcal{H}$ is an RKHS. A well-established approach using RKHS theory is to select $\hat{M}$ as a solution to the *regularized least squares problem*

$$\hat{M} = \arg\min_{M \in \mathcal{H}} \sum_{i=1}^{N} \|y_T^{(i)} - M(\boldsymbol{x}_T^{(i)})\|_{\mathbb{Y}_T}^2 + \gamma\|M\|_{\mathcal{H}}^2, \tag{11}$$

with $\gamma \in \mathbb{R}_+$ and $\|\cdot\|_{\mathcal{H}}$ a corresponding RKHS norm. As our target is a function-valued mapping $M(\cdot)$ – an operator – $\|\cdot\|_{\mathcal{H}}$ is induced by an *operator*-valued kernel $K: \mathbb{X}_T \times \mathbb{X}_T \mapsto \mathcal{B}(\mathbb{Y}_T)$ mapping to the space of bounded operators over the output space [53]. The salient feature of the above formulation (11) is its well-posedness: its solution exists and is unique for any $\mathcal{H}$, expressed as

$$\hat{M}(\cdot) = \sum_{i=1}^{N} K(\cdot, \boldsymbol{x}_T^{(i)})\beta_i, \quad \beta_i \in \mathbb{Y}_T \tag{12}$$

through a representer theorem [54]. Still, due to the Koopman-invariant structure (9) from Lemma 1, the choice of the RKHS $\mathcal{H}$ for $\hat{M}$ is not arbitrary. Thus, the question is how to craft $\mathcal{H}$ so the solution $\hat{M}$ is decomposable into Koopman operator eigenfunctions (9), forming an *LTI predictor*.

Firstly, it is obvious that (9) consists of summands that may lie in different RKHS, denoted as $\{\mathcal{H}^{\lambda_j}\}_{j=1}^{\bar{D}}$. Then, $\mathcal{H}$ is constructed from the following direct sum of Hilbert spaces [55]:

$$\tilde{\mathcal{H}} = \mathcal{H}^{\lambda_1} \oplus \cdots \oplus \mathcal{H}^{\lambda_{\bar{D}}} \ \text{ so that } \ \mathcal{H} = \text{range}(\mathcal{S}) := \{f_1 + \ldots + f_{\bar{D}} : f_1 \in \mathcal{H}^{\lambda_1}, \ldots, f_{\bar{D}} \in \mathcal{H}^{\lambda_{\bar{D}}}\} \tag{13}$$

with $\mathcal{S}: \tilde{\mathcal{H}} \to \mathcal{H}, (f_1 \cdots f_{\bar{D}}) \mapsto f_1 + \ldots + f_{\bar{D}}$ the summation operator [56]. Thus, to construct $\mathcal{H}$, a specification of the RKHS collection $\{\mathcal{H}^{\lambda_j}\}_{j=1}^{\bar{D}}$ is required, so that it represents Koopman eigenfunctions from (9).

**Theorem 1** (Koopman eigenfunction kernel)**.** *Consider trajectory data $\{\boldsymbol{x}_T^{(i)}\}_{i=1}^{N}$ from Assumption 1, a $\lambda \in \mathbb{C}$ and a universal (base) kernel $k: \mathbb{X} \times \mathbb{X} \mapsto \mathbb{R}$. Then, the kernel $K^\lambda: \mathbb{X}_T \times \mathbb{X}_T \mapsto \mathcal{B}(\mathbb{Y}_T)$*

$$K^\lambda(\boldsymbol{x}_T, \boldsymbol{x}_T') = \int_{\tau=0}^{T} \int_{\tau'=0}^{T} e^{-\lambda(\tau-t)} k\left(\boldsymbol{x}_T(\tau), \boldsymbol{x}_T'(\tau')\right) e^{-\lambda^*(\tau'-t)} \, d\tau d\tau', \tag{14}$$

(i) *defines an RKHS $\mathcal{H}^\lambda$,*

(ii) *is universal for every eigenfunction of Definition 1 corresponding to $\lambda$,*

(iii) *induces a data-dependent function space* $\text{span}\left\{K^\lambda(\cdot, \boldsymbol{x}_T^{(1)}), \ldots, K^\lambda(\cdot, \boldsymbol{x}_T^{(N)})\right\}$ *that is Koopman-invariant over trajectory-data $\{\boldsymbol{x}_T^{(i)}\}_{i=1}^{N}$.*

In Theorem 1, we derive an eigenfunction RKHS by defining its corresponding kernel that embeds the invariance transformation (7) over data samples. Also, we would like to highlight that the above result addresses a long-standing open challenge in the Koopman operator community [19–21], i.e., defining universal function spaces that are guaranteed to be Koopman-invariant. Now, we are ready to introduce the *Koopman kernel* as the kernel obtained by combining "eigen-RKHS" as described in (13).

**Proposition 1** (Koopman kernel)**.** *Consider trajectory data $\mathbb{D}_N$ of Assumption 1 and a set of kernels $\{K^{\lambda_j}\}_{j=0}^{\bar{D}}$ from Theorem 1. Then, the kernel $K: \mathbb{X}_T \times \mathbb{X}_T \mapsto \mathcal{B}(\mathbb{Y}_T)$ given by*

$$K(\boldsymbol{x}_T, \boldsymbol{x}_T') = \sum_{j=1}^{\bar{D}} K^{\lambda_j}(\boldsymbol{x}_T, \boldsymbol{x}_T') \tag{15}$$

(i) *defines an RKHS $\mathcal{H} := \mathcal{S}(\mathcal{H}^{\lambda_1} \oplus \cdots \oplus \mathcal{H}^{\lambda_{\bar{D}}})$,*

(ii) *is universal for any output (3b), provided a sufficient amount[4] of eigenspaces $\bar{D}$.*

Above, we have derived the "Koopman-RKHS" $\mathcal{H}$ for solving the problem (11) with a universal RKHS spanning Koopman eigenfunctions. Thus, the sample-data solution for an eigenfunction flow follows from the functional regression problem (11) and takes the form $\phi_{\lambda_j}(\cdot) = \sum_{i=1}^{N} K^{\lambda_j}(\cdot, \boldsymbol{x}_T^{(i)})\beta_i, \ \beta_i \in \mathbb{Y}_T$ — providing a basis for the LTI predictor.

---

[4]Sufficient amount is a rich enough set of eigenvalues $\{e^{\lambda_j[0,T]}\}_{j=1}^{\bar{D}}$ from $\overline{\mathbb{B}_1(\mathbf{0})}$ in $\mathbb{C}$ [57, Theorem 3.0.2].

## 3.2 Practicable LTI Predictor Regression

As a functional approximation problem, the solution of (11) is not parameterized by vector-valued coefficients, but rather functions of time. Although there are a few options to deal with function-valued solutions [53], we consider a vector-valued solution. A common drawback of such a discretization involves the loss of the inter-sample relations along the continuous signal. Crucially, this problem does not apply in our case, as the inter-sample relationships remain modeled for the discrete-time "Koopman kernel" due to its causal structure. Importantly, the vector-valued solution allows us to preserve all of the desirable properties derived in the continuous case.

Consider sampling $[0, T]$ at $H = T/\Delta t$ regular intervals to yield a discrete-time dataset from Assumption 1, discretized at points $\mathbb{H} \equiv \{t_0 \cdots t_H\}$. As a discretization of a function over time, with a slight abuse of notation, we denote the target vectors as $y_{\mathrm{H}} = [y(t_0) \cdots y(t_{\mathrm{H}})]^{\top}$. Thus, we are solving the time- and data-discretized version of the problem (5) that takes the form of a linear coregionalization model [58, 59].

**Corollary 1** (Time-discrete Koopman kernel)**.** *Consider trajectory data* $\{x_{\mathrm{H}}^{(i)}\}_{i=1}^{N}$ *and let* $\mu_j := \mathrm{e}^{\lambda_j \Delta t}$, $\boldsymbol{\mu}_j^{\top} := [\mu_j^0 \cdots \mu_j^H]$*. Then, the scalar-induced matrix kernel* $\boldsymbol{K}^{\mu_j} : \mathbb{X}_{\mathrm{H}} \times \mathbb{X}_{\mathrm{H}} \mapsto \mathcal{B}(\mathbb{Y}_{\mathrm{H}})$

$$\boldsymbol{K}^{\mu_j}(\boldsymbol{x}_{\mathrm{H}}, \boldsymbol{x}_{\mathrm{H}}') = \boldsymbol{\mu}_j \boldsymbol{\mu}_j^{*\top} \underbrace{\frac{1}{(H+1)^2} \sum_{m=0}^{H} \sum_{n=0}^{H} \mu_j^{-m} k^j\left(\boldsymbol{x}_{\mathrm{H}}(t_m), \boldsymbol{x}_{\mathrm{H}}'(t_n)\right) \mu_j^{*-n}}_{k^{\mu_j}(\boldsymbol{x}_{\mathrm{H}}, \boldsymbol{x}_{\mathrm{H}}')}, \qquad (16)$$

*satisfies the properties (i)–(iii) from Theorem 1 over* $\mathbb{H}$*, so that it defines an RKHS* $\mathcal{H}^{\mu_j}$*, is universal per Definition 1 over* $\mathbb{H}$ *with* $\mathrm{span}\{\boldsymbol{K}^{\mu}(\cdot, \boldsymbol{x}_{\mathrm{H}}^{(1)}), \ldots, \boldsymbol{K}^{\mu}(\cdot, \boldsymbol{x}_{\mathrm{H}}^{(N)})\}$ **(KI)** *over* $\{x_{\mathrm{H}}^{(i)}\}_{i=1}^{N}$*. Given a collection of kernels* $\{\boldsymbol{K}^{\mu_j}\}_{j=0}^{\bar{D}}$*, the matrix Koopman kernel* $\boldsymbol{K}^{\mu_j} : \mathbb{X}_{\mathrm{H}} \times \mathbb{X}_{\mathrm{H}} \mapsto \mathcal{B}(\mathbb{Y}_{\mathrm{H}})$

$$\boldsymbol{K}(\boldsymbol{x}_{\mathrm{H}}, \boldsymbol{x}_{\mathrm{H}}') = \sum_{j=1}^{\bar{D}} \boldsymbol{K}^{\mu_j}(\boldsymbol{x}_{\mathrm{H}}, \boldsymbol{x}_{\mathrm{H}}'), \qquad (17)$$

*satisfies the properties (i)–(ii) from Proposition 1 over* $\mathbb{H}$*, defining RKHS* $\mathcal{H}^{\Delta t} := \mathcal{S}(\mathcal{H}^{\mu_1} \oplus \cdots \oplus \mathcal{H}^{\mu_{\bar{D}}})$*.*

Now, we are fully equipped to obtain the time-discrete solution to our initial problem (5) provided a dataset of trajectories. Before presenting the solution to Koopman Kernel Regression, we introduce some helpful shorthand notation. We use the following kernel matrix abbreviations: $k_{\boldsymbol{X}\boldsymbol{X}} = [k(\boldsymbol{x}^{(a)}, \boldsymbol{x}^{(b)})]_{a,b=1}^{N}$, $k(\boldsymbol{x}, \boldsymbol{X}) = [k(\boldsymbol{x}, \boldsymbol{x}^{(b)})]_{b=1}^{N}$, $\boldsymbol{K}_{\boldsymbol{X}\boldsymbol{X}} = [\boldsymbol{K}(\boldsymbol{x}^{(a)}, \boldsymbol{x}^{(b)})]_{a,b=1}^{N}$ and $\boldsymbol{K}(\boldsymbol{x}, \boldsymbol{X}) = [\boldsymbol{K}(\boldsymbol{x}, \boldsymbol{x}^{(b)})]_{b=1}^{N}$.

**Proposition 2** (KKR)**.** *Consider a discrete-time dataset of Assumption 1,* $\mathbb{D}_N^{\Delta t} = \{x_{\mathrm{H}}^{(i)}, y_{\mathrm{H}}^{(i)}\}_{i=1}^{N}$*, and let* $\boldsymbol{y}_{\mathrm{H}}^{\top} = [y_{\mathrm{H}}^{(1)\top} \cdots y_{\mathrm{H}}^{(N)\top}]$ *with* $\otimes$ *the Kronecker product. Then,*

$$\boldsymbol{\alpha}_j = k_{\boldsymbol{X}_0 \boldsymbol{X}_0}^{-1} k_{\boldsymbol{X}_{\mathrm{H}} \boldsymbol{X}_{\mathrm{H}}}^{\mu_j} \left(\boldsymbol{I}_N \otimes \boldsymbol{\mu}_j^{*\top}\right) \underline{\boldsymbol{\beta}}, \qquad \underline{\boldsymbol{\beta}} = (\boldsymbol{K}_{\boldsymbol{X}_{\mathrm{H}} \boldsymbol{X}_{\mathrm{H}}} + \gamma \boldsymbol{I}_{H+1} \otimes \boldsymbol{I}_N)^{-1} \boldsymbol{y}_{\mathrm{H}} \qquad (18)$$

*defines a unique time-sampled solution to (11) in terms of eigenfunctions* $\hat{\phi}(\boldsymbol{x}_0) = [k_{\boldsymbol{x}_0 \boldsymbol{X}_0}^j \boldsymbol{\alpha}_j]_{j=1}^{\bar{D}}$*, determining an LTI predictor[5] with* $\boldsymbol{\Lambda} = \mathrm{diag}([\mu_1 \cdots \mu_D])$*,*

$$\boldsymbol{z}^+ = \boldsymbol{\Lambda} \boldsymbol{z}, \ \boldsymbol{z}_0 = \hat{\phi}(\boldsymbol{x}_0), \qquad (19a)$$

$$\hat{y} = \boldsymbol{1}^{\top} \boldsymbol{z}. \qquad (19b)$$

Notice how in (18), we re-scale the trajectory domain to that of the state-space. This enables us to write the forecast of (19), with a slight abuse of notation, using an extended observability matrix [60]

$$\hat{y}_{\mathrm{H}} = \boldsymbol{\Gamma} \hat{\phi}(\boldsymbol{x}_0), \qquad \boldsymbol{\Gamma} := \left[\boldsymbol{1}^{\top} \ \boldsymbol{1}^{\top} \boldsymbol{\Lambda} \ \cdots \ \boldsymbol{1}^{\top} \boldsymbol{\Lambda}^H\right]^{\top}. \qquad (20)$$

The confinement to a non-recurrent domain plays a crucial role in making the base kernel RKHSs isometric to "eigen-RKHSs" $\mathcal{H}^{k^j} \cong \mathcal{H}^{k^{\mu_j}}$ via *invariance transforms*, guaranteeing a feasible return from the time-series domain $\mathbb{X}_T$ to the state-space domain $\mathbb{X}_0 \subseteq \mathbb{X}$ for evaluating the model over initial conditions.

**Remark 1.** *The salient feature of our proposed KKR framework compared to existing methods is the fact that Koopman-invariance* **(KI)** *over data samples is independent from the outcome of an optimization algorithm, e.g. minimizing the forecasting risk to compute* $\underline{\boldsymbol{\beta}}$ *in (18). Thus, we are able to directly optimize for a downstream task (forecasting)* **(OR)** *given a suitably rich set of eigenvalues.*

---

[5]For discrete-time predictors, we omit the time-step specification and denote the next state with "$(\cdot)^+$".

### 3.3 Selecting Eigenvalues

Until now, we have used the sufficient cardinality $\bar{D} \in \mathbb{N}$ of an eigenvalue set that encloses [61] or is the true spectrum. However, we have provided no insight regarding the selection of $\bar{D}$ spectral components or how they can be estimated. Here, we go beyond the learning-independent and non-constructive existence result of [28] and provide a consistency guarantee and relate it to sampling eigenvalues without the knowledge of the true spectrum.

**Proposition 3.** *Consider the oracle Koopman kernel $\boldsymbol{K}(\boldsymbol{x}_\mathrm{H}, \boldsymbol{x}_\mathrm{H}')$ and a dense set $\{\mu_j\}_{j=1}^{\infty}$ in $\overline{\mathbb{B}_1(\boldsymbol{0})}$. Then, $\|\boldsymbol{K}(\boldsymbol{x}_\mathrm{H}, \boldsymbol{x}_\mathrm{H}') - \sum_{j=1}^{D} \boldsymbol{K}^{\mu_j}(\boldsymbol{x}_\mathrm{H}, \boldsymbol{x}_\mathrm{H}')\|_{\mathcal{B}(\mathbb{Y}_\mathrm{H})} \to 0, \forall\, \boldsymbol{x}_\mathrm{H}, \boldsymbol{x}_\mathrm{H}' \in \mathbb{X}_\mathrm{H} \text{ as } D \to \infty.$*

As shown in Proposition 3, even if we do not know the *oracle* kernel, we can arbitrarily approximate it by sampling from a dense set supported on the closed complex unit disk $\overline{\mathbb{B}_1(\boldsymbol{0})}$ [57, Theorem 3.0.2] with the error vanishing in the limit $D \to \infty$. There is no loss of generality when considering the unit disk as any finite radius disk can be scaled in the interval $[0, T]$. Furthermore, approximation of the oracle kernel by sampling a distribution over $\overline{\mathbb{B}_1(\boldsymbol{0})}$ leads to an almost sure $\mathcal{O}(1/\sqrt{D})$ convergence rate. It is conceivable that faster rates can be obtained in practice by including prior knowledge to shape the spectral distribution, e.g. using well-known concepts such as leverage-scores or subspace orthogonality [62, 63]. Based on spectral priors one can include a more biased sampling technique by precomputing components of the operator spectrum, e.g. computing Fourier averages [64], to determine the phases $\omega_j$ of complex-conjugate pairs $\mu_{j,\pm} = |\mu_j|\,\mathrm{e}^{\pm\mathrm{i}\omega_j}$ and sample the modulus from another physics-informed distribution. However, rigorous considerations of optimized and efficient sampling are beyond the scope of this paper and rather a topic of future work.

### 3.4 Numerical Algorithm and Time-Complexity

---
**Algorithm 1** Regression and LTI Forecasts using KKR

---

Data $\mathbb{D} = \{\boldsymbol{x}_\mathrm{H}^{(i)}, y_\mathrm{H}^{(i)}\}_{i=1}^{N}$, Eigenvalues $\{\mu_j\}_{j=1}^{D}$
**function** REGRESS($\mathbb{D}, \{\mu_j\}_{j=1}^{D}$)
    form Gramians $k_{\boldsymbol{X}_0\boldsymbol{X}_0}, \{k_{\boldsymbol{X}_\mathrm{H}\boldsymbol{X}_\mathrm{H}}^{\mu_j}\}_{j=1}^{D}, \boldsymbol{K}_{\boldsymbol{X}_\mathrm{H}\boldsymbol{X}_\mathrm{H}}$
    fit mode operator $\hat{M}(\cdot)\colon \mathbb{X}_\mathrm{H} \mapsto \mathbb{Y}_\mathrm{H}$ (18, right)
    recover eigenfunctions $\hat{\boldsymbol{\phi}}(\cdot)\colon \mathbb{X}_0 \mapsto \mathbb{Z}_0$ (18, left)
    construct $\boldsymbol{\Gamma}\colon \mathbb{Z}_0 \mapsto \mathbb{Y}_\mathrm{H}$ (20, right)
    **return** *LTI predictor* $\boldsymbol{\Gamma}\hat{\boldsymbol{\phi}}(\cdot)\colon \mathbb{X}_0 \mapsto \mathbb{Y}_\mathrm{H}$
**end function**
**function** FORECAST($\boldsymbol{x}_0$)
    "lift" $\boldsymbol{z}_0 = \hat{\boldsymbol{\phi}}(\boldsymbol{x}_0)$
    rollout $\hat{y}_H = \boldsymbol{\Gamma}\boldsymbol{z}_0$
    **return** trajectory $\hat{y}_H$
**end function**

---

For a better overview, the pseudocode for regression and forecasting of our method are shown in Algorithm 1. We also put the time-complexity of our algorithm into perspective w.r.t. Koopman operator regression of PCR/RRR [23] and ridge regression using state-of-the-art signature kernels [48] (RR-Sig-PDE) in Table 1. The training complexity of our KKR is comparable to that of RR-Sig-PDE regression and generally better than that of PCR/RRR. Given that accurate LTI forecasts require higher-rank predictors, the seemingly mild quadratic dependence makes $D^2 > NH$ and leads to a more costly matrix inversion. Furthermore, our LTI predictor also has a slightly better forecast complexity due to not depending on trajectory length. Obviously, due to a mere matrix multiplication after an initial nonlinear map, LTI predictors have a significantly lower evaluation complexity than the nonlinear predictor of Sig-PDE's. Due to requiring updated observation sequences as inputs, Sig-PDE kernels introduce a raw evaluation complexity that is also quadratic in sequence length.

| Method | Training | $H$-step forecast | | |
|---|---|---|---|---|
| KKR (ours) | $\mathcal{O}(N^3H^3 + DN^2H^2d)$ | $\mathcal{O}(DH + DNd)$ | $N$ | # trajectories |
| PCR/RRR | $\mathcal{O}(D^2N^2H^2 + N^2H^2d)$ | $\mathcal{O}(DH + DNHd)$ | $H$ | trajectory length |
| RR-Sig-PDE | $\mathcal{O}(N^3H^3 + N^2H^2l^2d)$ | $\mathcal{O}(NH^2l^2d)$ | $D$ | predictor rank |

| $l$ | # time-delays |
|---|---|
| $d$ | dim(input data) |

Table 1: Time complexities.

## 4 Learning Guarantees

With a completely defined KKR estimator, we assess its essential learning-theoretic properties, i.e., the behavior of the learned functions w.r.t. to the ground truth with increasing dataset size.

## 4.1 Consistency

Although well-established in most function approximation settings [65–67], the setting of Koopman-based LTI predictor learning for nonlinear systems is void of consistency guarantees. Here we use a definition of universal consistency from [68] that describes the uniform convergence of the learned function to the target function as the sample size goes to infinity for any compact input space $\mathbb{X}$ and every target function $q \in C(\mathbb{X})$. The existing convergence results for Koopman-based LTI predictors [69] are in the sense of strong operator topology — allowing the existence of empirical eigenvalues that are not guaranteed to be close to true ones even with increasing data [70]. This lack of spectral convergence has a cascaded effect in Koopman operator regression as, in turn, the convergence of eigenfunctions and mode coefficients is not guaranteed. Here, the convergence of modes is replaced by the convergence of eigenfunctions, and convergence of spectra is replaced by the convergence of (20) to the mode decomposition operator $\hat{M} \equiv \mathbf{\Gamma}\hat{\phi} \to M \equiv \mathbf{\Gamma}\phi$ with the estimate denoted by $\hat{(\cdot)}$.

**Theorem 2** (Universal consistency). *Consider a universal kernel $\mathbf{K}$ (17) and a data distribution supported on $\mathbb{X}_{\mathrm{H}} \times \mathbb{Y}_{\mathrm{H}}$. Then, as $N \to \infty$, $\|M - \hat{M}\|_{\mathbb{Y}_{\mathrm{H}}} \to 0$ and $\|\phi_{\mu_j} - \hat{\phi}_{\mu_j}\|_{\mathbb{Y}_{\mathrm{H}}} \to 0, \forall j = 1, \ldots, D$.*

## 4.2 Generalization Gap: Uniform Bounds

Due to formulating the LTI predictor learning problem as a function regression problem in an RKHS, we can utilize well-established concepts from statistical learning to provide bounds on the generalization capabilities of KKR. Given a dataset of trajectories, the following *empirical risk* is minimized

$$\hat{\mathcal{R}}_N(\hat{M}) := \tfrac{1}{N} \sum_{i \in [N]} \|y_{\mathrm{H}}^{(i)} - \hat{M}(\boldsymbol{x}_{\mathrm{H}}^{(i)})\|_{\mathbb{Y}_{\mathrm{H}}}^2$$

which is "in-sample" mean square error (MSE) w.r.t. a trajectory-data generating distribution $\rho_{\mathcal{D}}$ of i.i.d. initial conditions. The *true risk*/generalization error of an estimator is the "out-of-sample" MSE of the model on the entire domain and denoted as $\mathcal{R}(\cdot)$. Those quantities are, in essence, the model's performance on test and training data, respectively. Allowing for statements on the test performance with an increasing amount of data by means of training performance is a desirable feature in data-driven learning. Hence, we analyze our model in terms of the *generalization gap*

$$|\mathcal{R}(\hat{M}) - \hat{\mathcal{R}}_N(\hat{M})| = \left| \mathbb{E}_{(\boldsymbol{x}_{\mathrm{H}}, y_{\mathrm{H}}) \sim \rho_{\mathcal{D}}} [\|y_{\mathrm{H}} - \hat{M}(\boldsymbol{x}_{\mathrm{H}})\|_{\mathbb{Y}_{\mathrm{H}}}^2] - \tfrac{1}{N} \sum_{i=1}^N \|y_{\mathrm{H}}^{(i)} - \hat{M}(\boldsymbol{x}_{\mathrm{H}}^{(i)})\|_{\mathbb{Y}_{\mathrm{H}}}^2 \right|. \quad (21)$$

To ensure a well-specified problem, we require models in the hypothesis to admit a bounded norm.

**Assumption 2** (Bounded RKHS Norm). *The unknown function $M$ has a bounded norm in the RKHS $\mathcal{H}^{\Delta t}$ attached to the Koopman kernel $\mathbf{K}(\cdot, \cdot)$, i.e., $\|M\|_{\mathcal{H}^{\Delta t}} \leq B$ for some $B \in \mathbb{R}_+$.*

The above smoothness assumption is mild, e.g., satisfied by band-limited continuous trajectories [71] and computable from data [72, 73]. In stark contrast, well-specified Koopman operator regression [23] requires the operator to map the RKHS onto itself, which is a very strong assumption [34, 35].

To derive the main result of this section, we utilize the framework of Rademacher random variables for measuring complexity of our model's hypothesis space, a concept generally explored in [74] and more particularly for classes of operator-valued kernels in [75]. Conveniently, the derivation is, in terms of the RKHS $\mathcal{H}^{\Delta t}$, similar to standard methods on RKHS-based complexity bounds [74]. We use well-known results based on concentration inequalities to provide high probability bounds on a model's generalization gap in terms of those complexities. Finally, we upper bound any constant with quantities specified in our assumptions and can state the following result.

**Theorem 3** (Generalization Gap of KKR). *Let $\mathbb{D}_N^{\Delta t} = \{\boldsymbol{x}_{\mathrm{H}}^{(i)}, y_{\mathrm{H}}^{(i)}\}_{i=1}^N$ be a dataset as in Assumption 1 consistent with a Lipschitz system on a non-recurrent domain. Then the generalization gap (21) of a model $\hat{M}$ from Proposition 2 under Assumption 2 is, with probability $1 - \delta$, upper bounded by*

$$|\mathcal{R}(\hat{M}) - \hat{\mathcal{R}}_N(\hat{M})| \leq 4RB\sqrt{\frac{\kappa H^2}{N}} + \sqrt{\frac{8 \log \frac{2}{\delta}}{N}} \in \mathcal{O}\left(\frac{H}{\sqrt{N}}\right), \quad (22)$$

*where $R$ is an upper bound on the loss in the domain, and $\kappa$ the supremum of the base kernel.*

We observe an overall dependence of order $\mathcal{O}(1/\sqrt{N})$ w.r.t. data points, resembling the regular Monte Carlo rate to be expected when working with Rademacher complexities. Remarkably, an increase in the order of the predictor $D$ cannot widen the generalization gap but will eventually decrease the

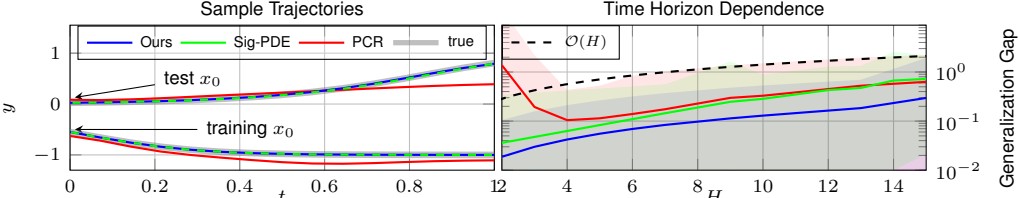

Figure 2: Forecasting performance (48 i.i.d. runs) for the bi-stable system for $H{=}14$ and $N{=}50$ for respectively optimal $D_{\text{KKR}}{=}100$, $D_{\text{PCR}}{=}10$ and 15 delays for Sig-PDEs. **Left:** Exemplary trajectories showing the advantage of learning with time-series kernels. **Right:** The generalization gap with an increasing forecast horizon, demonstrating generalization advantages of KKR.

empirical risk due to the consistency of eigenspaces (Proposition 3). Combined, our findings are a substantial improvement, both quantitatively and in terms of interpretability, over existing risk bounds on forecasting error [23, Theorem 1]. Additionally, our intuitive non-recurrence requirement is easily verifiable from data. In contrast, the Koopman operator regression in RKHS comes with various strong assumptions [35] that require commonly unavailable expert knowledge. Also, the generalization of existing Koopman-based statistical learning approaches depends on rank while ours is rank-independent. The significant implications of our results are demonstrated in the following.

## 5    Numerical Experiments

In our experiments[6], we report the squared error of the forecast vector for the length of data trajectories averaged over multiple repetitions with corresponding min-max intervals. We validate our theoretical guarantees and compare to state-of-the-art operator and time-series approaches in RKHS. For fairness, the same kernel and hyperparameters are chosen for our KKR, PCR (EDMD), RRR [23] and regression with signature kernels (Sig-PDE) [48]. Note, PCR and RRR are provided with the same trajectory data split into one-step data pairs while the time and observation time-delays are fed as data to the Sig-PDE regressor due to its recurrent structure. Along with code for reproduction of our experiments, we provide a JAX [76] reliant Python module implementing a sklearn [77] compliant KKR estimator at https://github.com/TUM-ITR/koopcore.

**Bi-stable system**  Consider an ODE $\dot{x} = ax + bx^3$ that arises in modeling of nonlinear friction. The parameters are $a = 4$, $b = -4$, making for a bi-stable system at fixed points $\pm 1$. The numerical results are depicted in Figure 2. Sample trajectories both on training and testing data indicate the utility of the forecast risk minimization of KKR. While EDMD correctly captures the initial trend of most trajectories it fails to match the accuracy of Sig-PDE or our KKR predictors that utilize time-series structure. Furthermore, the behavior of KKR's generalization gap for an increasing time horizon $T = H\Delta t, \Delta t = 1/14s$ closely matches our theoretical analysis.

**Van der Pol oscillator**  Consider an ODE $\ddot{x} = \dot{x}(2 - 10x^2) - 0.8x$ describing a dissipative system whose nonlinear damping induces a stable limit cycle — a phenomenon present in various dynamics.

In Figure 3 two fundamental effects are validated: the generalization gap with increasing data and consistency with test risk that does not deteriorate for increasing eigenspace cardinality. The performance of PCR/RRR is strongly tied to predictor rank while Sig-PDE's less so w.r.t. delay length.

Table 2: Average risk (20 runs) $[\times 10^{-2}]$ for Van der Pol for various *spectral sampling* and *lengthscales*, $N{=}200$, $H{=}14$.

| $\rho(\mu)$ | uniform | | boundary-biased | | physics-informed | |
|---|---|---|---|---|---|---|
| $D$ | 16 | 200 | 16 | 200 | 16 | 200 |
| $\mathcal{R}_{\ell=10^1}$ | 13.7 | **5.38** | 11.2 | **5.38** | 5.60 | 5.58 |
| $\mathcal{R}_{\ell=10^0}$ | 6.46 | **0.78** | 4.10 | **0.78** | 0.97 | 0.92 |
| $\mathcal{R}_{\ell=10^{-1}}$ | 7.12 | **1.74** | 4.33 | **1.74** | 1.83 | 1.80 |

Crucially, our KKR approach does not require a careful choice of the eigenspace cardinality to perform for a specific amount of data. Although the eigenvalues that determine the eigenspaces are randomly chosen from a uniform distribution in the unit ball, KKR consistently outperforms PCR/RRR. In Table 2 we show the spectral sampling and hyperparameter effects. We employ the following strategies: *uniform* - uniform distribution on the complex unit disk, *boundary-biased* - a distribution on the complex unit disk skewed towards the unit circle, *physics-informed* - eigenvalues of various vector field Jacobians. As expected, physics-informed performs well with lower rank

---

[6]Additional details on the numerical experiments can be found in the supplementary material.

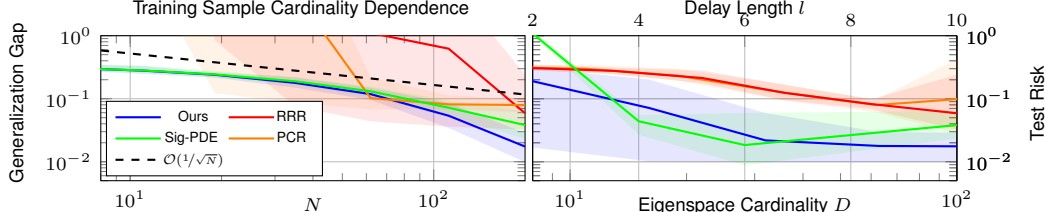

Figure 3: Forecasting risks (20 i.i.d. runs) for the Van der Pol system over a time-horizon $H = 14$ ($T = 1s$). **Left:** Generalization gap for the best $D$ / $l$ (ours 500, PCR 62, RRR 100, RR-Sig-PDE 10) is depicted with a growing number of data points. **Right:** Test risk behavior with an increasing amount of eigenspaces is shown for $N = 200$. Shaded areas depict min-max risk intervals.

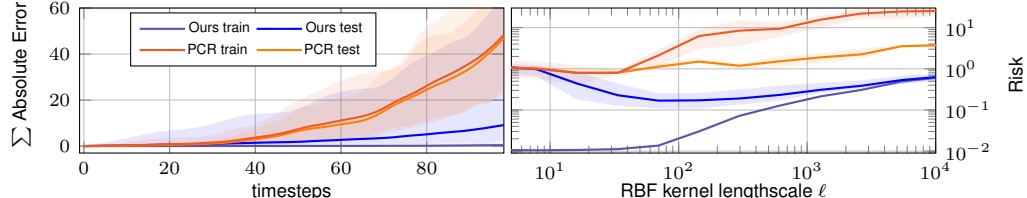

Figure 4: Cumulative error and forecast risks (5 train-test splits) for flow past cylinder data and $H = 99$. Our KKR with orders-of-magnitude greater usable $\ell$-range and accuracy. **Left:** Cumulative absolute error for the best $D/\ell$ (ours 200/70, PCR 200/35) is depicted over timesteps. **Right:** Forecast risk for 99 steps within a range of RBF lengthscales. Shaded areas depict min-max intervals.

compared to uninformed approaches. However, it is outperformed by unit-ball sampling approaches for higher rank due to a lack of coverage. Table 3 includes CPU timings for completeness.

| #data $= N \times H = 200 \times 14$ | KKR | PCR | RRR | Sig-PDE |
|---|---|---|---|---|
| Training [s]/Forecast [ms] | **8.0/ 54** | 90/ 84 | 88/150 | 8.6/5900 |

Table 3: Computation times for Van der Pol.

**Flow past a cylinder** We consider high-dimensional data of velocity magnitudes in a Kármán vortex street under varying initial cylinder placement, as illustrated in Figure 5.

The cylinder position is varied on a $7 \times 7$ grid in a $50 \times 100$-dimensional space and the flow is recorded over $H = 99$. The quantity of interest is a velocity magnitude sensor placed in the wake of the cylinder. In forecasting from an initial velocity field, KKR outperforms PCR by orders-of-magnitude as shown in Figure 4. We omit Sig-PDE regression due to persistent divergence after $\approx 20$ steps. The latter is hardly surprising, given that Sig-PDE models iterate one step predictions based only on the shapes of time-delays while LTI models directly output time-series from initial conditions.

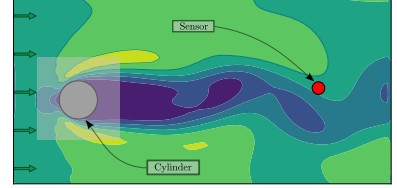

Figure 5: Flow illustration. Area of initial cylinder positions shaded.

## 6   Conclusion

We presented a novel statistical learning framework for learning LTI predictors using trajectories of a dynamical system. The method is rooted in the derivation of a novel RKHS over trajectories, which solely consists of universal functions that have LTI dynamics. Equivalences with function regression in RKHS allow us to provide consistency guarantees not present in previous literature. Another key contribution is a novel rank-independent generalization bound for i.i.d. sampled trajectories that directly describes forecasting performance. The significant implications of the proposed approach are confirmed in experiments, leading to superior performance compared to Koopman operator and sequential data predictors in RKHS. In this work, we confined our forecasts to a non-recurrent domain for a specific length of trajectory data, where the choice of spectra is arbitrary. However, exploring more efficacious spectral sampling schemes is a natural next step for extending our results to asymptotic regimes that include, e.g., periodic and quasi-periodic behavior. It has to be noted that vector-valued kernel methods have limited scalability with a growing number of training data and output dimensionality. Therefore, exploring solutions that improve scalability is an important topic for future work. Furthermore, to enable the use of LTI predictors in safety-critical domains, the quantification of the forecasting error is essential. Hence, deriving uniform prediction error bounds for KKR is of great interest.

**Acknowledgements**

The authors acknowledge the financial support of the EU Horizon 2020 research and innovation programme "SeaClear" (ID 871295) and the ERC Consolidator grant "CO-MAN" (ID 864686). Petar Bevanda also thanks Jan Brüdigam for feedback, and Vladimir Kostić and Pietro Novelli for useful discussions on operator regression.

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
