# OpenReview forum: "Koopman Kernel Regression"
_NeurIPS.cc/2023/Conference — NeurIPS 2023 poster_

### Official Review · Reviewer_oAtn · 2023-07-04

**Soundness:** 2 fair
**Presentation:** 2 fair
**Contribution:** 2 fair
**Rating:** 4
**Confidence:** 3

**Summary:**

This paper proposes a Koopman eigenfunction kernel that induces a Koopman-invariant RKHS and the corresponding analytical solution of the regression problem with this kernel. They further provide theoretical guarantees and numerical experiments to demonstrate the efficacy of the proposed method.

**Strengths:**

[1] Designing the kernel that induces a Koopman-invariant RKHS is a challenging problem, and this paper contributes to this problem.

[2] The theoretical results in Section 4 guarantee the performance of the proposed method.

**Weaknesses:**

[1] The presentation is poor, for example, there should be more clearly mathematical description about the Koopman approximation theory in RKHS and the claimed open challenge in the Koopman operator community; in Line 13, a noun is missing after 'aforementioned'.

[2] The authors only test the proposed method in low-dimensional toy models.

**Questions:**

[1] In Lemma 2, how to guarantee the Koopman eigenfunctions span the continuous functions space $C(\mathbb{X})$, or you cannot obtain the universality of eigenfunctions?

[2] The way of omitting spatial modes in Eq. (5) is not standard, and even it can simplify the formulation in your mathematical deduction, the spatial modes are still required in numerical experiments to approximate any measurement function $q$. How do you represent any measurement $y_T=q(x_T)$ without coefficients in Eq. (5)?

[3] In Lemma 1, the authors define $g$ as $g:\mathbb{X}_0\to C(\mathbb{X}_0)$, in this way $g$ is a functional not a function. However, $\phi_\lambda$ is the Koopman eigenfunction related to the original dynamics $f$ and $g$ is any functional, how do you guarantee the Eq. (4) stand? If the Lemma 1 was wrong, the Koopman-invariant property in Theorem 1 (iii) cannot stand either.

[4] Why not compare your method with the kernel based Koopman operator learning methods such as [R1-3]? The EDMD and PCR are classic methods but their performance is now lagging behind.

__References__

[R1] Baddoo, P. J., Herrmann, B., McKeon, B. J., & Brunton, S. L. (2022). Kernel learning for robust dynamic mode decomposition: linear and nonlinear disambiguation optimization. Proceedings of the Royal Society A, 478(2260), 20210830.

[R2] Kostic, V., Novelli, P., Maurer, A., Ciliberto, C., Rosasco, L., & Pontil, M. (2022). Learning dynamical systems via koopman operator regression in reproducing kernel hilbert spaces. Advances in Neural Information Processing Systems, 35, 4017-4031.

[R3] Meanti, G., Chatalic, A., Kostic, V. R., Novelli, P., Pontil, M., & Rosasco, L. (2023). Estimating Koopman operators with sketching to provably learn large scale dynamical systems. arXiv preprint arXiv:2306.04520.

**Limitations:**

[1] In reality, the Koopman eigenvalues and eigenfunctions are unknown and are exactly the learning objectives.  However, this paper requires the Koopman eigenvalues to construct the Koopman kernel, which is impractical in real-world scenarios.

[2] The authors only test their method in toy models instead of real-world dataset such as PIV data for flow over a cylinder or molecular dynamics.

---

> ### Author Rebuttal · Authors · 2023-08-06
>
> First of all, we wish to thank the reviewer for their insight and the catching a notational oversight.
> ### **On weaknesses**
> 1.  We thank the reviewer for their insight. To clarify, we do not address the problem of learning Koopman operators but LTI predictors and try to convey that in Section 2.  Although popular, EDMD-like approaches are by no means the only way to obtain LTI predictors. We take a function approximation approach and thus present non-DMD method (at least not in the classical sense) for learning LTI predictors. We will expand on this difference in the updated paper and support it by a corresponding optimization objective.
> 2. We thank the reviewer for voicing their concerns but we believe that the flow-over-cylinder experiment from Appendix C is no toy example. There, our KKR-based
> LTI predictor performs well in a sparse data regime on a high-dimensional dataset.
> ### **On limitations**
> 1.  We definitely see how this could be true for much of the existing work. However,  we stress that, **our work has no requirement of knowing the spectra, which is confirmed in both theory and experiments**. Why this is true traces back to operator theoretic results we rely on. On a non-recurrent domain any and all eigenvalues are legitimate [1] while the eigenfunctions are so rich that they are universal approximators [2]. When learning, different outputs of interest bring out different eigenfunctions for the same eigenvalues - just like different modes correspond to different measurement functions in the mode analysis of an estimated Koopman operator. Finally, our consistency results prescribe a finite amount of eigenfunctions that leads to arbitrarily accurate forecasting while sampling from any distribution with support on the closed unit disk is the recipe to achieve the latter. Importantly, our experiments are randomized in a way that no single run had the same set of eigenvalues, confirming that knowing or learning the spectra is not required.
> 2. We thank the reviewer for voicing their concern. However, **there is nothing to suggest that KKR only works on toy examples based both on theory and experiments**, cf. Appendix C of the supplemental. There, KKR performs quite well in a sparse data regime when constructing an LTI predictor in a flow-over-cylinder setting.
> ### **Answers to questions**
> 1. The Lemma 2 of the Appendix A is an auxiliary result that connects existing results on non-recurrence and operator theory that include universality. For our approach, non-recurrence and a universal base kernel provide the ingredients for obtaining universality, cf. Proposition 1, (ii). Also, in Section 3.3 of our work we provide a "recipe" for constructing a universal RKHS using Koopman eigenfunction RKHSs.
> 2. Indeed, omitting spatial modes is not standard but possible w.l.o.g. in our setting. Contrary to Koopman operator learning approaches, our KKR directly learns the eigenfunctions for a measurement function. Therefore, the modes are absorbed by individual eigenfunctions so that a normal form, e.g., modes equal to one, can be subsumed. Although we remark on this in footnote of Lemma 2, we will include the reasoning in the updated main text.
> 3. We are very thankful to the reviewer for catching this unfortunate typo in our manuscript. It was never our intent to introduce such an ambiguity. We want to clarify that we do not use or require functionals for the maps $q(\cdot),\boldsymbol{g}(\cdot)$ or $\boldsymbol{\phi}(\cdot)$ and related theoretical results. Accordingly, all occurrences of the expression "$:\mathbb{X}_0 \rightarrow C\left(\mathbb{X}_0\right)$" are going to be replaced with "$\in C\left(\mathbb{X}_0\right)$'' as functions are what we consider. **Thus, the correctness of our results is unaffected.**
> 4. We appreciate bringing works [R1-3] to our attention. When it comes to problem setting and scope, the work in [R1] is unrelated to our approach. The reasoning is threefold: 1) no LTI predictor learning, 2) missing a statistical learning framework, 3) lack of any statistical learning guarantees. \
> Actually, we compare our own to the results of [R2] both theoretically and empirically. From the two estimators (PCR and RRR) that are shown in [R2] we implemented one, as the paper showed that they are quite similar in terms of performance. Due to the reviewer's request, we will add the RRR estimator to our comparison. The new comparison (cf. global response PDF) with RRR fully confirms our intuition as our KKR estimator clearly outperforms RRR by a similar margin observed for PCR (EDMD). \
> Finally, there was no possibility of being aware of [R3] as the arXiv preprint appeared weeks after our submission. Although improving the scalability of PCR and RRR, the forecasting performance is bounded by that of PCR and RRR estimators [R2] for the same data. Therefore, we have decided to not include this method for comparison in our paper, but instead show the results for ridge regression using state-of-the-art *signature kernels* for time series modeling as suggested by the reviewer \#GSvu.
>
> [1] Bollt, E. M. (2021). Geometric considerations of a good dictionary for Koopman analysis of dynamical systems: Cardinality, “primary eigenfunction,” and efficient representation. Communications in Nonlinear Science and Numerical Simulation, 100. \
> [2] Korda, M., & Mezic, I. (2020). Optimal Construction of Koopman Eigenfunctions for Prediction and Control. IEEE TAC, 65(12), 5114–5129.

---

> > ### Comment · Reviewer_oAtn · 2023-08-19
> > **Review to the rebuttal**
> >
> > Dear authors,
> >
> > Thanks for the detailed rebuttal. The additional experiments have solved my concerns. I am happy to amend my score of the paper.

---

### Official Review · Reviewer_GSvu · 2023-07-04

**Soundness:** 2 fair
**Presentation:** 2 fair
**Contribution:** 3 good
**Rating:** 5
**Confidence:** 4

**Summary:**

This papers introduces a, to my knoweldge, new kernel on trajectories for dynamical system learning derived from Koopman operator theory. Consistency guarantees and generalisation bounds are derived using classical RKHS arguments. Numerical experiments on Bi-stable system and Van der Pol oscillator are performed using the proposed technique.

**Strengths:**

The idea of making links between Koopman operator theory and kernel methods is interesting. The proposed kernel is to my knowledge new.

**Weaknesses:**

There is a general lack of mathematical rigour throughout the paper (see my remarks and questions in the next section).  Some statements are unjustified and at times untrue. For example in line 49-50 it is claimed that the RKHS associated to the derived kernel is the first in the literature that posseding universal features that have linear dynamics. *Signature kernels* [1] have such property and are universal for the much broader class of continuous functions on (rough) paths. Furthermore, the numerical examples are very simplistic and the experimental details are poorely described.

The paper has good potential. I am willing to increase my rating if the authors are able to address my points below reagarding the mathematical rigour and provide a fair theoretical and numerical comparison with [1].

**References**

[1] Salvi, Cristopher, et al. "The Signature Kernel is the solution of a Goursat PDE." SIAM Journal on Mathematics of Data Science 3.3 (2021): 873-899.

**Questions:**

- For readers unfamiliar with the theory of dynamical systems it is worth reminding the definition of forward-complete system in line 74, perhaps as a footnote.
- Equation (1b) is slightly ambiguous as the function $q$ acts pointwise on the path $x: t \mapsto x(t)$; it would be clearer to write $y(t) = q(x(t))$ for any $t \in \mathbb{T}$.
- It is worth specifying what are the minimal regularity conditions required from $f$ and $q$ in (1a) and (1b) so that the flow in line 76 exists and is unique.
- The map $g$ in equation (2a) is vaguely introduced as a "feature map"; can the authors provide more details regarding this map?
- I think combining equations (3a) and (3b) into a single equation would make the optimisation objective clearer.
- In lines 87-88 it is claimed that (3) *is generally not analytically solvable or tractable*; this is not true in general, as (3) is nothing but an ODE and numerous methods for fitting parameters of ODEs are available, including the recent Neural ODE framework [1] and many others.
-  In lines 137-138 it is mentioned that "the Koopman eigen-functions in Definition 1 are universal approximators of continuous functions"; can the authors give more details about this statement? What's the hypothesis space of continuous functions mentioned in this sentence?
- Lemma 1 is vague and not rigorously stated: what is the space $\mathbb X_0$? Is $C(\mathbb X_0)$ the space of continuous functions from $\mathbb X_0$ to $\mathbb R$? What's the regularity of $g$? To talk about continuity one needs a topology on input and output spaces; what are the choice of topologies in this Lemma? What does the wording "... are assigned..." in line 146 mean precisely? The function $M$ implicitly depends on $c,g,A$ so I don't understand the claim of the Lemma. The norm $|| \cdot ||_{\mathbb Y_T}$ has not been been defined.
- Point (ii) in Theorem 1: what do you mean by universal here?
- It seems to me that the kernel defined in equation (10) does not take its values on the space $\mathcal{B}(\mathbb Y_T)$ of bounded linear operators on $\mathbb Y_T$.
- Is the kernel defined in Proposition 1 positive (semi)definite?
- Theorem 2 needs to be written in much higher level of details, recalling the definitions of $M, \mathbf \Gamma, \hat \phi$.
- The link to the github repository in footnote 6 breaks the anonymity of submission rule as it reveals the identity of the some of the authors.

**References**

[1] Chen, Ricky TQ, et al. "Neural ordinary differential equations." Advances in neural information processing systems 31 (2018).

**Limitations:**

The authors adquately discuss limitations and future work.

---

> ### Author Rebuttal · Authors · 2023-08-07
>
> We thank the reviewer on the extensive constructive feedback that certainly improves our paper.
> ### **On weaknesses**
> We respectfully disagree with the reviewer on unjustified and untrue statements as we elaborate below.
> ##### **Difference to signature kernels**
> - Although the provided reference [1] suggests a valuable kernel for sequence modeling, the Koopman and signature kernels solve different problems (Koopman eigenfunctions vs Goursat PDEs), have different properties and are applied in different contexts (ODEs vs controlled DEs [2]). Our KKR approach exploits the Koopman kernel as means to an LTI state-space model determined by its initial condition. In contrast, when used for forecasts, the signature kernel processes a sequence of past observations driving the predictor - leading to a more local yet nonlinear model. Which model type should be preferred is, of course, debatable and ultimately depends on the downstream task. For optimization-based decision, the computational simplicity of forecasting is paramount. In that case, our LTI state-space model is advantageous as it reduces forecasting to matrix multiplication or a linear constraint.
> Following the reviewer’s suggestion, **we support the above theoretical comparison by experiments (cf. global response and accompanying PDF).**
> ##### **Linear dynamics & universality**
> - Signature kernels have linear dynamics in the restrictive case when sequences themselves are linear in time [1, Sec. 3.1]. For nonlinear dynamics, the suggested Sig-PDE kernel has piece-wise linear dynamics.
> In contrast, the Koopman kernel is universal and its dynamics linear, regardless of whether the time-series are generated by a linear or nonlinear dynamical system. We will briefly clarify this in the updated version.
> #### **Simplicity and lack of experimental detail**
> - It seems that we miss a reference to Appendix C of the supplemental in the main text. In the updated version, we will point to it in a footnote. We believe the additional details and experiments therein (including a flow-over-cylinder dataset) can alleviate the reviewer's concerns.
> ### **Answers to questions**
> - For the sake of brevity we hoped to cover it by delegating to a reference but try to build this into the main text.
> - We believe the suggested change is possibly misleading at that instant in the paper. The continuous-time model has no explicit time dependence and separates into an ODE and an measurement function - ubiquitous in dynamical systems modeling [3]. Thus, we would prefer to keep our notation.
> - We require Lipschitz continuous maps $\boldsymbol{f}$ and ${q}$. We clarify this in the updated version.
> - We intended to state this vaguely and set up an abstraction layer to discuss various methods in the literature. This feature/``lifting'' map is an important aspect in learning LTI predictors, and the literature is filled with different choices of this mapping. Finding a Koopman-invariant (KI) choice for it that fits the system at hand is the open challenge we are aiming to address.
> - We can understand how this might be tempting but believe that it would compromise clarity. Satisfying (3b) is the very property that existing works on constructing LTI models are unable to guarantee.
> - The fact that (3) is a constrained functional regression problem does not lead to it being generally tractable, let alone analytically solvable. Approximations need to be put in place to make it tractable using discretization schemes, e.g., time-discretization or space-discretization. We do not introduce such discretizations immediately, which allows us to preserve the core structure of (3), such as Koopman-invariance. We will expand on this aspect in the updated version.
> -  It is considers $\mathbb{C}$-valued continuous functions over a compact set equipped with the sup-norm. Due to spatial constraints, we refer to our supplemental (Appendix A) that includes auxiliary results.
> -  The set $\mathbb{X}_0$ is introduced on line 81 and it represents initial conditions that form a non-recurrent domain. It suffices to take $g \in C(\mathbb{X}_0)$ of complex-valued continuous functions equipped with the sup-norm. This will be added in the notation paragraph.
> We thank the reviewer for pointing out a lack of clarity, we will restate Lemma 1 as follows: "Under the flow of (1a), a function $g(\cdot)$ is transformed into a Koopman eigenfunction of Definition 1 via...".
>     * TL;DR for the claim of the lemma: (1) given a sufficiently rich amount of eigenvalues, $g$'s can be replaced by eigenfunctions, (2) by extension $\boldsymbol{A}$ is diagonal, (3)  $\boldsymbol{c} =\boldsymbol{1}$ w.l.o.g. We will explicitly require the sufficient richness in the updated version. The norm corresponds an $L^2$ norm, as implied by the definition of the output space.
> - We mean that any open eigenfunction of Definition 1 can be approximated to arbitrary precision using this kernel.
> - This is due to the regression goal being an operator $M$ which is a bounded operator over the output domain, well-established in operator-valued regression [64-66]. We will include the reasoning in the updated paper.
> - Indeed it is. Considering conjugate pairs of eigenvalues (possible w.l.o.g.), it becomes clear that the kernel is real-valued and positive definite. By polarizing their integrands, it follows that is p.s.d. as the integrals and sums of kernels preserve the base kernel's definiteness. It is probably interesting to note here that even without the usage of conjugate pairs, we did not observe any issues related to non-positive definiteness in our experiments.
> -  We will provide more details in the updated paper.
> -  Footnote 6 provides a publicly available link to the code accompanying [24] so **there is no breach of anonymity**.
>
> [2] P.K. Friz et al., Multidimensional stochastic processes as rough paths: theory and applications, Cambridge University Press, 2010.\
> [3] H.K. Khalil, Nonlinear Systems, Pearson Education, 2002.

---

> > ### Comment · Reviewer_GSvu · 2023-08-12
> >
> > I thank the authors for their responses. The additional comparisions and experiments in particular are very good! They should be incorporated into the main text should the paper be accepted. Nonetheless, it appears that the authors' responses have not addressed most of my suggestions. Signature kernels have linear dynamics no matter what the driving path is! The signature solves a linear CDE and the corresponding Goursat PDE for the kernel is also a linear hyperbolic PDE, even if the driving paths are not linear. This should clarified. I do not agree with the statement *leading to a more local yet nonlinear model*. In the attached pdf the forecasting time for Sig-PDE seems very high compared to all others, I wonder if that's a misprint. The concept of a feature map should be clearly defined. Examples of feature maps should be specified. The fact that the paper is only 8 pages can't be used as an argument for the lack of mathematical rigour in the writeup (even more so as the machine learning audience is becoming more and more math savvy). All in all, I keep my rating unchanged for now. I will make a final decision after consultation with other reviewers and AC.

---

> > > ### Author Response · Authors · 2023-08-15
> > >
> > > We thank the reviewer for their response and the recognition of our comparisons and experiments. We will definitely incorporate the new results!
> > > #### **What is meant by *linear dynamics***
> > > The linear dynamics we consider follow the classical definition of linear time-invariant (LTI) ODEs. In the notation of [1], an LTI predictor forecast
> > > $$
> > > y_t = \boldsymbol{c}^\top \mathrm{e}^{\boldsymbol{A} t}{\boldsymbol{z}}_0
> > > $$
> > > results from a solution of a homogenous LTI ODE.
> > > The state-transition matrix $\boldsymbol{A}$ has no dependence on any parameters and the ODE flow depends only on $\boldsymbol{z}_0$. Our Koopman kernel enforces the above LTI dynamics to hold by construction and KKR learns the mapping $\boldsymbol{z}_0$ so that it satisfies the above  predictor model at discrete points in time.
> > >
> > > Although we agree with the reviewer on linear CDE dynamics, such dynamics are still dependent on driving paths and do not amount to an (autonomous) LTI ODE. Thus, it represents a concept that is fundamentally different from the one we use in our work. \
> > > Appendix B.4 of [1] on solving *linear CDEs* provides a model whose dynamics are dependent on driving path changes $dx_{s_i}$
> > >
> > >  $$
> > >  y_t=y_0+\sum_{k=1}^D(\underset{0<s_1<\ldots<s_k<t}{\int \ldots \int} A(d x_{s_1}) \ldots A(d x_{s_k})) y_0
> > >  $$
> > >  analogous to a linear parameter/time-variant model in the classical ODE sense - cf. Peano-Baker series [2]. The driving parameters update from one to another prediction step and prohibit a forecast akin to the one for LTI ODE dynamics.
> > >  > Signature kernels have linear dynamics no matter what the driving path is!
> > >
> > > Note that, to have LTI ODE dynamics, a signature kernel's dynamics
> > >
> > > $$
> > > \frac{\partial^2 k_{x, x^\prime}}{\partial s \partial t}=c(s,t) k_{x, x^\prime}
> > > $$
> > >
> > > need to have *the path inner product* $c(s,t) \equiv \langle\dot{x}_s, \dot{x}^\prime_t\rangle_V$ constant (independent of any variables) holding jointly for *all driving paths over their entire time-intervals*. The latter is not the case for signature kernels as $c(s,t)$ is only constant on local subdomains of the paths' time-intervals [1, Sec. 3.1].\
> > > For the sake of clarity, we will add an identifier to our statement, e.g.,  "universal kernel with LTI dynamics" in the updated paper. \
> > > As mentioned in our rebuttal, we will revise the presentation and relation to existing work to include aspects of the above elaboration.
> > > #### **More local yet nonlinear model**
> > > > I do not agree with the statement leading to a more local yet nonlinear model.
> > > >
> > > For reference, the prediction model with Sig-PDE kernels reads
> > > $$
> > > {y^+}= f(x_t)= [k_{sig} (x_t,x^1_s),\dots, k_{sig} (x_t,x^n_s)]\boldsymbol{\alpha}
> > > $$
> > > with forecasts a sequential application of the above with updated controls ${x}_{t}$, following [1, Sec. 5.2].
> > >
> > > The locality of the Sig-PDE models is attributed to the requirement of being driven by past observations (changing the local dynamics of the system) unlike for ODEs that solely depend on an initial condition. The nonlinearity/complexity is obvious as the Sig-PDE kernels do not form LTI predictors in [1] and thus have higher forecasting complexity.
> > >
> > > #### **Regression with signature kernels has the highest forecast complexity**
> > > >  In the attached pdf the forecasting time for Sig-PDE seems very high compared to all others, I wonder if that's a misprint.
> > >
> > > As we mentioned in our global response, we used the kernels as in the code accompanying [1] but with no optimization for computing the signatures. Our complexity analysis (rebuttal PDF, Table 1) indicates that the high forecast time should not be surprising as Sig-PDE:
> > > -  has a raw evaluation complexity that is *quadratic* in the length of the input sequence
> > > -  regression model computes signatures w.r.t. *all of the data* at *every step of the forecast*
> > >
> > > LTI predictors have an inherent complexity advantage due to requiring only a single evaluation of kernel sections and a matrix multiplication for forecasting.
> > > #### **Feature maps**
> > > Following the reviewer's suggestion, we will formally define the concept of feature maps as well as discuss different ones found in the literature.
> > > #### **On addressing the reviewer's suggestions**
> > > We would like to point out that we addressed all of the reviewers points and suggestions in the rebuttal. Due to the limited space in the response, it is conceivable that we did not address them to the highest level of detail.
> > > We are happy to clarify any and all points to alleviate the reviewer's concerns.
> > >
> > > [1] Salvi, C., et al. "The Signature Kernel is the solution of a Goursat PDE." SIAM Journal on Mathematics of Data Science 3.3 (2021): 873-899.\
> > > [2] Baake, M.; Schlaegel, U. "The Peano Baker Series". Proceedings of the Steklov Institute of Mathematics. 275 (2011): 155–159.

---

### Official Review · Reviewer_xMDr · 2023-07-05

**Soundness:** 3 good
**Presentation:** 2 fair
**Contribution:** 3 good
**Rating:** 6
**Confidence:** 3

**Summary:**

The presented manuscript suggests a novel algorithm for learning dynamical systems trajectories called Koopman Kernel Regression. This method based on the Koopman operator theory proposes a unified way to construct observables such that the transformed dynamical system becomes LTI. Moreover, this study presents convergence results and the generalization risk bounds for the KKR method. The experiments confirm the given theory and demonstrate the performance of the KKR over existing kernel-based methods to learn the Koopman operator based on the observed snapshots.

**Strengths:**

1) convergence results and generalization risk bounds are developed and supported by experimental results
2) also experiments illustrate the more accurate and robust w.r.t. prediction horizon compared to EDMD
3) the introductory part with a description of the Koopman operator approach and the corresponding challenges is well-written and clearly states the objective of this study
4) kernel for Koopman eigenfunction is derived and theoretically analyzed

**Weaknesses:**

1) the key difference with the EDMD approach is not explicitly stated in the text, only the issues of the latter
2) it is unclear how to compose the complete algorithm with data samples as input and prediction as output from the presented theoretical results. The readability and accents in sections 3.2-3.3 should be fixed (from theory to practical usage)
3) how to select the kernel $k$ is not discussed in sufficient detail, only required properties are mentioned.
4) the discussion of the sampling strategy's importance for the obtained good experimental results is missing. In addition, how this step affects the performance of the KRR for arbitrary input data is interesting.


**Questions:**

1) what is a particular form of the universal kernel $k$ used in experiments? Do EDMD and KKR use the same kernel inside?
2) what about the computational complexity of the proposed algorithm? How does it scale with the increasing dimension of the ODE that generates training trajectories? Experiments use only 1d trajectories.
3) please provide a complete step-by-step description of the proposed method in the algorithm LaTeX environment. It is hard to extract the final algorithm steps from the presented list of theorems and remarks.
4) the experiment section should be extended to more challenging dynamical systems like the Lorenz system, and systems with high-dimensional states. The runtime comparison with EDMD is very interesting, too.
5) please provide a more detailed description of the experimental setup. In particular, what spectrum sampling strategy is used and why? What happened if one will use another strategy and how to select the best one?

**Limitations:**

No limitations are provided in the manuscript.

---

> ### Author Rebuttal · Authors · 2023-08-09
>
> We thank the reviewer for their constructive comments.
> ### **On weaknesses**
> 1. It seems that the decision to guide the reader through the limitations of existing work, in the hope of better motivating our own approach, was less fortunate. Admittedly, we realize how the explicit difference to the operator regression approach of EDMD might thus get lost. We want to highlight that our KKR is a function regression method that uses an RKHS comprised of Koopman eigenspaces to fit an output of interest. Conversely, EDMD-like methods regress a matrix surrogate of an infinite-dimensional operator and subsequently project it on an output of interest. In the updated paper, we will formally support the above elaboration with a corresponding optimization objective.
> 2. We understand that our focus on learning-theoretic results can be disadvantageous in understanding how to use our method in practice. Based on the reviewer's comment, we will provide the pseudocode for our algorithm. Additionally, we will also provide ``scikit-learn`` compatible open-source code with the updated paper.
> 3. It seems that we miss a reference to Appendix C of the supplemental in the main text. In the updated version, we will point to it in a footnote. We believe the additional details and experiments therein can alleviate the reviewer's concerns. Based on reviewer's feedback, we now also provide an extended study of kernel hyperparameters and sampling distributions in the global response comment and PDF.
> 4. Given the evident importance, we will address these aspects more clearly in the updated version of the main paper. We now also show the experimental performance of different sampling distributions that are biased towards specific regions of the unit disc, demonstrating their effect on the learning outcome. The figures supporting these experiments can be found in the global response and accompanying PDF.
> ### **On limitations**
> An explicit limitations section in the main paper turned out to be infeasible due to spatial constraints. We stress, however, that the conclusion section very much includes the limitations of our approach (also acknowledged by reviewer #GSvu).
>
> ### **Answers to questions**
>
> 1. We sided with the popular choice of the squared exponential kernel. Indeed, the same kernel and kernel hyperparamters are used for both methods. More details are found in the supplemental, Appendix C.
> 2. We appreciate the reviewer's relevant question on complexity. Therefore, we now provide training and forecasting complexities of our KKR method together with a comparison to Koopman operator as well as time-series modeling state-of-the-art is provided in the global response and the accompanying PDF. \
> We would like to point out that in the Van der Pol experiment we consider a vector-valued observable which amount to solving 2 regressions. Also, the content of Appendix C demonstrates that our method performs well in a sparse data regime for high-dimensional fluid flow data.
> 3. We welcome this suggestion. As mentioned, we will try to fit an algorithm in the form of pseudocode for the updated version.
> 4. Based on the reviewer's request, we conducted a run-time comparison to EDMD in the global rebuttal response (attached PDF, Table 2). \
> Regarding challenging dynamics, the Appendix C of the supplemental already provides additional experiments. In particular, we demonstrate that our KKR approach performs well in a sparse data regime on a high-dimensional dataset in flow-over-cylinder setting.
> 5. In our statistical experiments we sampled from a uniform distribution over the closed unit disk to not introduce any prior knowledge or preconceptions.  As any eigenvalues from the closed unit-disc are legitimate (based on Koopman operator theory), we sampled from a uniform distribution over the closed unit disk in our statistical experiments. The idea was that such a "flat" distribution would not introduce any prior knowledge or preconceptions. Also, the experiments were randomized such that no single run had the same set of eigenvalues, confirming the theoretical results, which prescribe that the more of them we take, the better it gets. \
> Based on the reviewer's comment, in the attached single-page PDF of the global response, we now assess different sampling distributions - demonstrating their effect on the learning outcome. \
> We appreciate raising the interesting question on optimal spectral sampling. As part of future work, we are currently investigating the problem of optimal spectral sampling.

---

> > ### Comment · Reviewer_xMDr · 2023-08-19
> >
> > Dear authors,
> > Thanks for the detailed response! The evaluation score remains the same.

---

### Official Review · Reviewer_aayt · 2023-07-10

**Soundness:** 4 excellent
**Presentation:** 4 excellent
**Contribution:** 3 good
**Rating:** 7
**Confidence:** 3

**Summary:**

Learning time-series obeying nonlinear dynamics is essential for forecasting. Often Koopman operator theory is used to replace the non-linear dynamics with a linear system where the state evolution is given by a high-dimensional matrix that has to be estimated/learned. In this paper, the authors bring the kernel methods together with Koopman operators to obtain a mathematical framework where learning guarantees can be studied. Eigenfunctions of the Koopman operator associated to a dynamical system can be approximated by the basis functions of a Reproducing Kernel Hilbert Space (RKHS) and learning these eigenfunctions reduces to functional kernel regression. The paper derives risk bounds using statistical learning theory and demonstrate their method with experiments.

**Strengths:**

The paper is well-written with a clear presentation. The problem of Koopman linearization is well-defined and techniques introduced are novel in terms of applying kernel methods for function approximation in this framework. Also, supplementary material is well-written and very helpful!

Theorem 1 clearly defines the operator-valued Koopman Kernel for a single eigenfunction in terms of a chosen base kernel (e.g. Gaussian RBF) and the rank of the full Koopman kernel is obtained by direct-summation of $\bar D$ such RKHS's. The main problem in this formalism is the choice of eigenvalues for the Koopman Kernel. The authors propose a method for sampling eigenvalues, while it is not clear how to me (See weaknesses).

**Weaknesses:**

1. In my opinion, previous methods are not explained well, and it is not easy to understand how this method differs from the previous works.

2. It could be helpful to comment on the spectral biases of the Koopman Kernels based on the base kernel and also the choice of Koopman eigenvalues. To me, it is still not clear how $\mu_i$'s are chosen, and it is essential to understand that point because Koopman kernel requires both the choice of base kernel and the spectrum.

3. Experiments might be extended to cover how the kernel hyperparameters affect the regression risk. For example, it is well-known in kernel methods community that certain spectral decay is essential for generalization. Since there are two spectra involved in this case (spectrum of the base kernel and spectrum of Koopman operator), it would be interesting to see how both affect generalization.

**Questions:**

1. What is the computational complexity/cost of this method, and how does it relate to the existing methods?

**Limitations:**

Limitations section is missing.

---

> ### Author Rebuttal · Authors · 2023-08-09
>
> We thank the reviewer for their recognition of our work.
> ### **On weaknesses**
> 1. Guiding the reader through the limitations of existing work to motivate our own approach seems to have been a less fortunate decision. Admittedly, we see how the explicit difference to the operator regression approach of EDMD might get lost in that. We will elaborate more on the fundamental difference between direct eigenfunction learning or KKR compared to operator learning (e.g. EDMD) in the updated paper.
> Specifically: Our KKR approach is not a DMD method, as there is no operator regression involved. KKR is a function regression method that uses an RKHS comprised of Koopman eigenspaces to fit an output of interest. Conversely,  EDMD-like methods regress a matrix surrogate of an infinite-dimensional operator and only subsequently project it on an output of interest. We will formally support the above elaboration with a corresponding optimization objective.
> 2. Tracing back to operator theoretic results we rely on, it is sufficient to sample a distribution with support on the closed unit disk. On a non-recurrent domain, any and all eigenvalues are legitimate while the eigenfunctions are so rich that they are universal approximators. Additionally, our consistency results prescribe a finite amount of eigenfunctions that leads to arbitrarily accurate forecasting while sampling from any distribution with support on the closed unit disk is the recipe to achieve that. In addressing the reviewer's concerns, we now
> also show the experimental performance of different sampling distributions that are
> biased towards specific regions of the unit disc, demonstrating the effect different
> sampling distributions have on the learning outcome. The figures supporting these
> experiments can be found in the the global response PDF.
> 3. We appreciate this interesting remark. Based on the reviewer's request we extend our experiments with a study of the effects different hyperparameters and spectral sampling schemes have on KKR's generalization (attached PDF in the global response, Figure 2).
>
> ### **On limitations**
> An explicit limitations section in the main paper turned out to be infeasible due to spatial constraints. We stress, however, that the conclusion section very much includes the limitations of our approach (also acknowledged by reviewer \#GSvu).
>
> ### **Answers to questions**
> 1. We appreciate the reviewer's question. Therefore, we now provide training and forecasting complexities of our KKR method.
> The complete comparison to Koopman operator time-series regression state-of-the-art is provided in the global response and the attached PDF therein.
>    - TL;DR: The methods are not directly comparable, due to their different solution approaches: in different regimes different methods would prevail.
> However, when constructing higher-rank predictors - required for forecast accuracy - our KKR method has a more favorable time-complexity than EDMD/RRR (linear vs quadratic in rank) and generally slightly better forecast complexity.

---

> > ### Comment · Reviewer_aayt · 2023-08-21
> >
> > I thank the authors for further clarifications.

---

### Author Rebuttal · Authors · 2023-08-08

We thank the reviewers for taking the time to review our manuscript and providing constructive comments that certainly improve our manuscript. We have addressed the reviewers’ comments and will update our paper accordingly. Please find below a joint response to some common concerns of the reviewers as well as the content description of the attached single-page PDF. Please let us know if there are any further suggestions. We look forward to hearing back from you!

We would like to point out that there is a lot of useful information already available in our supplemental that includes, but is not limited to:
   - background on (Koopman) operator theory - Appendix A
   - proofs of theoretical results - Appendix B
   - implementation details and additional experiments (e.g., including high-dimensional fluid flow data) - Appendix C \
 In the updated paper we plan to also add a statistical comparison to other methods (see additional experiments below) in the flow-over-cylinder experiment.
### **Difference to existing work**
We want to highlight that we do not directly address the problem of learning Koopman operators or time-series modeling. While both can be used for forecasting, they ought to be analyzed with rather different tools. Our work **exploits time-series structure and Koopman operator theory to provably learn *LTI predictors***. \
Although popular and supported to statistical learning concepts, Koopman operator learning (e.g. EDMD) is by no means the only way to address the problem of *statistical learning of LTI predictors*. One of our primary goals was to **provide a novel approach for statistical learning of LTI predictors** *rooted in function (not operator) approximation* theory that comes **with stronger theoretical guarantees** than existing work. The practical significance of these guarantees is demonstrated in our experimental evaluation.\
We stress that under the condition of non-recurrence, **knowing or learning Koopman operator eigenvalues is not required** for forecasting performance using LTI predictors - *which is confirmed both in theory and experiments*.

### **Additional results in the attached PDF**
- Following the requests of reviewers #GSvu and #oAtn, we include:
   -  comparison to RRR and Sig-PDE kernel regression in a forecasting task (attached PDF, Figure 1)
- Based on the suggestions of reviewers #aayt and #xMDr, we provide:
   - complexity analysis in comparison to existing work (attached PDF, Table 1)
   - runtime comparison (attached PDF, Table 2)
   - an empirical study of the effects  different hyperparameters and spectral sampling schemes have on KKR's generalization (attached PDF: Figure 2, Table 3)

#### **Forecast risk comparison to RRR and RR-Sig-PDE (Figure 1)**
For a fair comparison using the suggested signature kernels, we ran ``scikit-learn`` Ridge Regression (RR) using Sig-PDE kernels
https://github.com/crispitagorico/sigkernel. All methods used the same base kernel and the same lengthscales. For fairness w.r.t. RR-Sig-PDE (due to requiring time-delays), $l$-step backward simulation is done from $N$ i.i.d. trajectories of length $H$ used for PCR, RRR and KKR. Shown in Figure 1 of the attachment.

#### **Time-complexity (Table 1)**
Based on the reviewers' suggestions we include:
- *KKR train complexity:*$~\mathcal{O}(\text{inverse})+\mathcal{O}(\text{creation})=\mathcal{O}(\text{inverse})+D(\mathcal{O}(\text{Grammian})+\mathcal{O}(\text{Kronecker})+\mathcal{O}(\text{sum}))=\mathcal{O}(N^3H^3+D(N^2H^2d+N^2H^2+N^2H^2)) \in \mathcal{O}(N^3H^3+DN^2H^2d)$
- *KKR H-step forecast complexity:*$~\mathcal{O}(\text{matrix product})+\mathcal{O}(\text{kernel sections})=\mathcal{O}(DH)+\mathcal{O}(DNd)$

The complete comparison to PCR/RRR and RR-Sig-PDE is found in Table 1 of the attached PDF. As the comparison methods do not use multi-step/forecasting targets, we introduced $n=NH$ to make classical complexities for regression using PCR/RRR and RR + Sig-PDE kernels estimators recognizable.\
Our training complexity is comparable to RR-Sig-PDE and not directly comparable to PCR/RRR due to different solution approaches.
However, when constructing higher-rank predictors - often required for forecast accuracy - our KKR method scales more favorably than PCR/RRR (linear vs quadratic in $D$) with slightly better forecast complexity. The aforementioned LTI predictors scale favorably against RR-Sig-PDE's iterated nonlinear mapping evaluation due to a data-independent matrix multiplication after an initial nonlinear mapping.

#### **Runtime comparison (Table 2)**
We include these results based on the suggestion of reviewer \#xMDr. However, the results are not entirely comparable due to our KKR model being implemented in JAX https://github.com/google/jax. We adapt PCR/RRR from https://github.com/csml-iit-ucl/kooplearn and Sig-PDE kernels from https://github.com/crispitagorico/sigkernel. Note that, in the short rebuttal time, we did not manage to have the other methods running on the GPU or with precompiled code.
#### **Spectral sampling and hyperparameters (Figure 2, Table 3)**
In Figure 2 and Table 3 we show the influence of different base kernel lengthscales as well as spectral sampling strategies on generalization. \
For lengthscale selection, it is clear that a very small lengthscale does not generalize well quickly due to very slow spectral decay, as expected for kernel-based methods. \
Also confirming our theoretical results, as more eigenvalues are sampled, there is no significant difference observed between spectral sampling strategies and the only effect left influencing generalization is the spectral decay of the base kernel.

#### We will release ``scikit-learn``-compatible open-source code for the final submission, if the paper gets accepted.

#### We thank the reviewers and AC for their time and would happily provide additional clarification on our rebuttal as well as answer any further questions.

---

### Decision · Program_Chairs · 2023-09-21

**Decision:**

Accept (poster)

**Comment:**

Authors propose a new kernel derived from Koopman operator theory. Compared to prior methods based on Koopman operators, e.g. EDMD, it focuses on making a linear predictor. The paper provides generalization risk bounds, which are supported by experiments. However, the reviewers pointed out that the writing, assumption, and details on the experiments are missing. I strongly encourage the authors to make the final manuscript complete.